# Features of Allostatic Load in Patients with Essential Hypertension without Metabolic Syndrome Depending on the Nature of Nighttime Decreases in Blood Pressure

**DOI:** 10.3390/diagnostics13233553

**Published:** 2023-11-28

**Authors:** Tatyana Zotova, Anastasia Lukanina, Mikhail Blagonravov, Veronika Tyurina, Vyacheslav Goryachev, Anna Bryk, Anastasia Sklifasovskaya, Anastasia Kurlaeva

**Affiliations:** Institute of Medicine, RUDN University, 6 Miklukho-Maklaya St., 117198 Moscow, Russia; zotova-tyu@rudn.ru (T.Z.); lukanina-aa@rudn.ru (A.L.); 1042210206@rudn.ru (V.T.); goryachev-va@rudn.ru (V.G.); bryk-aa@rudn.ru (A.B.); sklifasovskaya-ap@rudn.ru (A.S.); kurlaeva-ao@rudn.ru (A.K.)

**Keywords:** essential hypertension, ABPM, allostasis, allostatic load, biological rhythm

## Abstract

Changes in the activity of the renin–angiotensin–aldosterone system are responsible for a stable shift in the regulation of the cardiovascular system in essential hypertension (EH). They can be characterized as hemodynamic allostasis. The purpose of our study was to determine the role of hemodynamic parameters in allostatic load in patients with EH without metabolic syndrome. Twenty-four hours of ambulatory blood pressure monitoring was performed, followed by linear and non-linear rhythm analysis. Based on the daily index, patients with EH were divided into two groups: group 1—patients with no significant nighttime decrease in blood pressure (BP); group 2—patients who had a nocturnal decrease in BP. The control group included healthy persons aged 25 to 69 years. A linear analysis was used to determine the mean values of systolic and diastolic BP, heart rate (HR), time load of BP, circadian index, and structural point of BP. Non-linear analysis was applied to determine the mesor, amplitude, range of oscillations and % rhythm of BP and HR. The allostatic load index (ALI) was also calculated on the basis of the corresponding biomarkers. It was found that ALI was significantly higher in groups 1 and 2 in comparison with the control group. The hemodynamic mechanisms of this increase were different.

## 1. Introduction

The phenotypic feature of essential hypertension (EH) involves different profiles of nighttime decreases in blood pressure (BP) due to an enhancement of the activity of the renin–angiotensin–aldosterone system (RAAS) at night [1,2]. The degree of a nocturnal BP decline in ambulatory blood pressure monitoring (ABPM) is determined on the basis of the daily index (DI). In accordance with the results obtained on the basis of DI, patients are commonly divided into the following types: dippers, non-dippers, over-dippers, and night-peakers [3,4]. In our study, a simpler gradation was applied: the patients were distributed into two groups depending on the presence or absence of a significant nighttime reduction in BP. DI ≥ 10% was considered a significant nocturnal decrease. Pathogenetic pharmacotherapy of EH decreases hemodynamic load on the cardiovascular system and reduces the risks of complications [5,6,7,8,9,10]. The adequacy of therapy is associated with the achievement of target levels of BP [11]. However, it is still poorly understood whether hemodynamic allostasis and allostatic load on the body are preserved or whether pathogenetic therapy returns regulatory mechanisms to homeostasis. At the same time, in the state of hemodynamic allostasis, the body cannot only exist under the conditions of altered parameters in the case of EH, but it also has some mechanisms that maintain this alteration resulted from the activation of the RAAS proteins, changes in the vegetative status, and the central mechanisms of regulation of vascular tone due to stress-induced reactions [1,2,12,13,14,15,16]. It is obvious that the main pathogenetic mechanism of allostatic load for patients with no significant nighttime decrease in BP can be a change in the daily dynamics of BP within the altered limits. At first glance, there are no mechanisms providing allostatic load in patients without a nocturnal BP decline. In this regard, the objective of our study was to determine the role of the hemodynamic component in the formation of allostatic load in patients suffering from EH with no metabolic syndrome by comparing the indicators reflecting hemodynamic allostasis with the values of the allostatic load index (ALI). It’s important that the study involved patients receiving antihypertensive therapy that reached the target values. Taking into account the fact that the development of allostasis is commonly accompanied by aging of the body, this problem requires a more thorough investigation [17,18,19,20,21,22].

## 2. Materials and Methods

The study included 102 participants. Patients with EH underwent outpatient examination and treatment at Outpatient Department of Moscow City Clinical Hospital No. 13. Informed consent was obtained from each participant. The study was approved by the Ethics Committee of the RUDN Institute of Medicine (code number: 10; date: 20 June 2019). Patients were diagnosed and staged on the basis of the Antihypertensive League’s Algorithms of Management for Patients with Arterial Hypertension, St Petersburg, 2019. All the patients received antihypertensive therapy according to the 2018/2021 recommendations of the European Society of Cardiology (ESC) and the European Society of Arterial Hypertension (ESH) working group on the treatment of arterial hypertension. All investigations were performed using certified equipment designed for this type of work (devices for 24 h of ambulatory blood pressure monitoring (ABPM) TM-2430 (A&D, Tokio, Japan), verified by Federal Budget Institution “Rostest—Moscow”. ABPM data were processed using ChronosFit (Heidelberg, Germany) [23,24] and EZDoctor 2.7 software. BP measurements were taken with intervals every 15 min from 8:00 am to 10:00 pm and every 30 min from 10:00 p.m. to 8:00 a.m. Night sleep time was determined using patients’ records. The compliance and validity of the study itself were over 85% successful.

### 2.1. Clinical and Laboratory Characteristics of Groups of Patients

Patients with co-morbidities that could alter the course of EH were not included in the study. Exclusion criteria were as follows: concomitant pathology that could change the course of EH (secondary hypertension, CAD, and metabolic syndrome). Metabolic syndrome was excluded using waist circumference (WC), body mass index (BMI), glycated hemoglobin (HbA1c), and immunoreactive insulin (IRI). Simpson ejection fraction (EF) was evaluated to exclude heart failure. Blood levels of creatinine and urea, as well as microalbuminuria (MAU), were used to exclude renal failure. Second-stage hypertension was diagnosed in all the patients (inclusion criterion). The profile of a nighttime decrease in BP was determined on the basis of DI. The patients with EH were divided into 2 groups depending on the presence or absence of a nocturnal BP decrease: group 1 (*n* = 32)—patients with EH with no significant nighttime decrease in BP (DI ≤ 10%); group 2 (*n* = 40)—patients with EH who had a nocturnal decrease in BP (DI ≥ 10%). All the patients with EH received adequate antihypertensive therapy to reach the target values (Table 1). Two groups of drugs were mainly used. The patients in group 1 received diuretics and calcium antagonists (aimed at reducing nocturnal BP). To determine normal hemodynamic parameters, a control group was included in the study: healthy persons aged 25 to 69 years (*n* = 30); 60% of them were males. When forming the control group, special attention was paid to the age range of the participants in order to take into account age-related changes in the cardiovascular system. The selection of participants into this group was carried out on the basis of compliance with normal indicators of average hemodynamic values and integrative indicators such as structural point of BP (SPBP), double product (DP), and circadian index (CI)—formulas for their calculation are given below.

### 2.2. Linear and Non-Linear Methods of ABPM Analysis

Linear analysis was used to determine the following indices: 24 h mean values of BP and heart rate (HR), daytime mean values of BP and HR, nighttime mean values of BP and HR, and time load of BP. Integrative indicators were also calculated: CI for systolic, diastolic BP (SBP and DBP) and HR; DP = HR × SBP/100; SPBP = DBP/SBP. Assessment criteria for DP were as follows—average value: from 76 to 89 conventional units; below average: ≥90 conventional units; above average: ≤75 conventional units. Assessment criteria for SPBP: this value reflects tissue perfusion conditions and is close to the “golden ratio” (0.618) in healthy subjects [25,26,27,28,29].

Time index for SBP and DBP is percentage of time when BP exceeds 140/90 mm Hg during the day and 120/80 mm Hg at night for a 24 h period.

Non-linear analysis was performed using the ChronosFit software, version 1.06 [23] to evaluate chronobiological features of 24 h profiles of BP and HR. Non-linear analysis is a combination of partial Fourier analysis with stepwise regression. To assess the rhythmic component in the regulation of the cardiovascular system, the following indicators were determined: mesor—average level of the indicator for a 24 h period; amplitude—maximum deviation of the corresponding indicator from the mesor for a 24 h period; range of oscillations—the difference between the maximum and minimum values of the indicator; % rhythm (power of oscillations)—index reflecting the proportion of the indicator values having oscillatory distribution for a 24 h period. The feasibility of using non-linear analysis is explained by its sensitivity, which is associated with the fact that the inverse Fourier transform restores the continuity of the 24 h dynamics of BP and HR.

Diagnosis of hemodynamic allostasis was carried out using data assessment of linear and non-linear analyses that were compared with normal values of the indicators in the control group. The possibility of mean values (M ± m) to analyze the obtained data is explained by the fact that hemodynamic parameters are the subject of stable regulation of the cardiovascular system even under hemodynamic allostasis.

### 2.3. Allostatic Load Index (ALI)

The basis for calculating the ALI for each patient is an assessment of quartile deviations for the following biomarkers: SBP, DBP, BMI, blood levels of low-density lipoproteins (LDL), high-density lipoproteins (HDL), triglycerides (TG), glucose, HbA1c, creatinine, albumin, fibrinogen, IRI, and using an appropriate calculator [5,6,30]. Statistical analysis of the obtained results was carried out using BioStat LE 7.3.0 software (AnalystSoft, Inc., Alexandria, VA, USA) The values obtained for each biomarker were used to determine the 25th and 75th percentiles. For all indices except HDL, values below the 75th percentile were coded as 0 and values above the 75th percentile were coded as 1; for HDL, values below the 25th percentile were coded as 1 and values above the 75th percentile were coded as 0. The coding values assigned to the biomarkers were summed. The resulting sum was equal to ALI. ALI rank was determined according to the following scale: 0—no allostatic load; 1–2—low allostatic load; 3–4—moderate allostatic load; 5 and above—high allostatic load.

### 2.4. Statistics

All data obtained were analyzed for the presence of statistically significant differences at *p* ≤ 0.05 using non-parametric Mann–Whitney *U*-test and Fisher angular transformation for percentages (one-way test). When comparing hemodynamic indices, the homogeneity of the groups was assessed based on the coefficient of variation (CV) of the index (standard deviation/mean value of the index); when its value was less than 30%, the group was considered homogeneous. This coefficient was used to perform further comparative analysis between groups. As the hemodynamic parameters are the subject of steady regulation even under conditions of hemodynamic allostasis, the statistical analysis of the obtained data was carried out using the mean values and their errors (M ± m).

## 3. Results

### 3.1. Clinical and Laboratory Characteristics of the Groups

A comparative analysis was carried out between groups 1 and 2 and between each of the groups and the control group sequentially. The analysis revealed the signs of hemodynamic allostasis, accompanied by changes in the nature of metabolism and tissue perfusion (Table 2). Intergroup comparison made it possible to identify hemodynamic features of hemodynamic allostasis and allostatic load for each group.

Comparative analysis of the groups, represented in Table 1, revealed no differences in metabolic changes, cardiac contractility, microalbuminuria, or age. But there were gender differences. There was a balance between males and females in group 1, while group 2 was dominated by females. There were no qualitative differences in drug therapy between the two groups.

### 3.2. Drug Therapy for Groups 1 and 2

The study of EH pathogenesis resulted in the development of several groups of effective antihypertensive agents, defined as pathogenetic treatments. In our research, we tried to find out whether the mechanisms regulating the activity of the cardiovascular system return to homeostasis or whether this regulation is carried out within the frame of hemodynamic allostasis, leading to an allostatic load on the body. In our study, all the patients received adequate antihypertensive treatment that reached target values. All the patients were treated with drugs from two groups (Table 1). Patients from group 1 more frequently received β-blockers and calcium antagonists. These medicines were used to correct a nighttime decrease in BP (calcium antagonists) and suppress sympathicotonia (β-blockers).

### 3.3. Results of Linear and Non-Linear Analysis

According to the results of the linear analysis of 24 h ABPM records, the groups were homogeneous in terms of hemodynamic parameters (BP and HR): CV did not exceed 30% (Table 2).

The values of nighttime SBP and DBP, as well as the CI for SBP and DBP, were significantly different between groups 1 and 2 (Table 2). The latter finding was due to the features of group formation. The time index for SBP and DBP in both groups was statistically different from the control. In this regard, we considered the activity of the cardiovascular system in patients with EH under conditions of altered BP indicators as a state of hemodynamic allostasis.

In order to clarify the mechanisms of its formation, we carried out non-linear analysis of ABPM data and determined the following indicators: mesor, amplitude, % rhythm (power of oscillations), and range of oscillations (Table 3). Both groups of patients with EH were characterized by a statistically significant increase in the values of the mesor of SBP and DBP, a decrease in % rhythm (Figure 1), and an increase in the proportion of ultradian rhythms compared to the control group. There was also a significant increase in amplitude of BP and HR as well as a decrease in range of oscillations for SBP in groups 1 and 2 in comparison with controls (Figure 2). Moreover, for the first group of patients, the hemodynamic load is formed by the absence of a nighttime decrease in BP determined on the basis of CI (Table 2). The amplitude of BP in this group of patients was reduced. This change provided a significant increase in mesor of SBP.

The revealed hemodynamic changes in both groups of patients were accompanied by differences from the control group in DP and SPBP (Table 2). The value of DP indicated a reduced rate of metabolism in groups 1 and 2 (99.5 ± 2.40 and 97.0 ± 1.9 vs. 89.60 ± 0.27). SPBP deviated from the “golden ratio” level (0.54 ± 0.08 and 0.57 ± 0.07 vs. 0.62 ± 0.002 in controls).

### 3.4. Assessment of ALI

Since the formation of allostatic load in cases of EH is closely related to stress [31,32,33], we calculated ALI [13,34,35,36,37,38,39] using the following biomarkers: SBP, DBP, BMI, blood levels of LDL, HDL, TG, glucose, glycated Hb, creatinine, urea, and immunoreactive insulin (IRI), applying quartile scoring. These data were compared with hemodynamic features of EH realization in patients with the presence/absence of an adequate nighttime BP reduction. The results are represented in Table 4.

It must be emphasized that groups 1 and 2 were significantly different from the control group in the value of ALI (Table 4). However, no statistically significant differences between groups 1 and 2 were found for this index. A comparative analysis of the percentage of patients in both groups with altered hemodynamic parameters and their comparison with each other revealed the leading hemodynamic mechanisms of allostatic load depending on the nature of a nighttime BP decrease. In particular, group 1 (no nighttime BP lowering) was characterized by an increase in the number of patients with HR over 80 bpm and a simultaneous decrease in CI (below 1) for all three parameters (SBP, DBP, and HR) in 28.1% of patients.

The obtained result indirectly confirms the involvement of the sympathoadrenal system [1] in altering the 24 h profile of BP and HR in group 1. Group 2 (patients with a nighttime decrease in BP) showed an increase in the percentage of patients with an extended range of SBP and DBP compared with group 1. In our opinion, the low allostatic load (ALI = 1–2) in both groups is due to the antihypertensive therapy. Moderate allostatic load (ALI = 3–4) was characteristic of 9.3% of patients in group 1 and in 5% of patients in group 2. This level of exertion is average and requires a review of the tactics of antihypertensive therapy.

It is also important to note that the identified hemodynamic changes in both groups were accompanied by pronounced dynamics of integrative indices, reflecting the level of metabolism (DP) and tissue perfusion conditions (SPBP). For the latter one, there was a marked deviation from the “golden ratio” that was characteristic of the control group. The level of metabolism in both groups, respectively, was assessed as low compared to the average in the control group (Table 3).

## 4. Discussion

For the first time, the possibility of allostatic load on the body with adverse effects was reported by B.S. McEwen and E. Stellar in their publications in 1993 [40,41,42,43]. According to a number of authors, allostatic load is defined as the severity of allostasis, which manifests as a more pronounced response than necessary. Some authors associate the possibility of allostasis development with stress effects on the body [13,32]. Stress itself and its forms (acute or chronic) are directly related to the characteristics, duration, and consequences of exposure to persistently altered neurohumoral regulation [1,2,13]. It is the assessment of the consequences of these effects on the body (hemodynamics and metabolism) that led to the possibility of tabulating these effects based on ALI [21,37,38]. This approach turned out to be quite fruitful in occupational medicine, sports, and psychology [16,38,44]. In relation to this problem, various biomarkers of allostatic load are used, depending on the mechanisms of fixation of altered regulation. In particular, this load can result both from physiological adaptation [39,45] and stress-related pathological conditions [21,46]. However, this approach is rather rarely used in clinical practice [47].

Since EH can be classified as a stress-induced disease [33], in this study, we compared specific hemodynamic changes in patients with EH (considered hemodynamic allostasis), with ALI counted on the basis of the following parameters: SBP, DBP, BMI, blood levels of LDL, HDL, TG, glucose, glycated Hb, IRI, creatinine, urea, IRI, and MAU. Each participant in the study was assigned a score based on the quartile assessment. The absence of metabolic syndrome in the patients in groups 1 and 2 allowed us to relate the existing increase in ALI to the specific hemodynamic changes for each group. In the patients in group 1, there is a change in the 24 h dynamics of BP visualized on the basis of CI (below 1). These changes in the circadian profile of BP are also confirmed by a reduced percentage of the 24 h rhythm with the predominance of ultradian rhythms [4]. This profile of the circadian rhythm is characteristic of disorders of water-salt metabolism [48,49,50,51]. Analysis of ALI in this group revealed an increase in the number of patients with increased HR (over 80 bpm) in addition to the changes in CI. In this regard, we consider these features as hemodynamic allostasis, which is realized through the connection between the activity of the RAAS proteins and the sympathoadrenal system [52].

An increase in allostatic load in patients with an impaired nighttime profile of BP is a predictable result of this study. However, the increase in ALI up to 1–2 in group 2 could not be predicted in advance. In addition to the increase in the percentage of ultradian rhythms of BP, an expansion of amplitude and range of oscillations for BP should be noted among the features of hemodynamics in this group (Table 3). We associate the formation of a time load for BP in this group, particularly with these changes. The significance of the established hemodynamic changes in patients with EH is confirmed by the dynamics of BP and SPBP (Table 2). These indices are integral and confirm the fact that the implementation of the activity of the cardiovascular system under adequate antihypertensive therapy is accompanied by a stable metabolic shift and altered tissue perfusion [53,54,55]. This fact suggests a role for mitochondria in the formation of hemodynamic allostasis in EH [38,56,57,58,59]. However, there is still no resolution of the question concerning the primacy of intracellular metabolic changes due to the development of heterogeneous pools of mitochondria [57,60] and systemic regulatory changes (RAAS and autonomic dysfunction) realized under stress in patients with EH. In our opinion, some changes in regulatory influences may occur when the body adapts to the new conditions of existence. But these changes would be just functional (reversible) [45]. We believe that constancy in the shift of regulatory influences and the formation of allostasis are only possible in cases of stable changes in cellular metabolism due to the appearance of a heterogenic pool of mitochondria.

It should be noted that the concept of hemodynamic allostasis as a factor of allostatic load was used by the authors for the first time when analyzing hemodynamics in pregnant women with EH, and it is associated with hypervolemia during pregnancy [61]. Previous studies have also established that allostatic load in patients with EH [19] without metabolic syndrome does not depend on patients’ age and can be considered a general characteristic of EH [4]. Taking into account the pathogenetic links between allostatic load and aging [17,18,20,21,37,39,58,62], the increasing number of patients with EH [8,9,63], the possibility of using allostasis indicators to assess the effectiveness of therapy seems to be an urgent issue. The results of our study indicate that focusing only on target levels of BP does not provide an opportunity to reduce allostatic load.

The formation of an allostatic load on the body is directly related to the mode of control over central hemodynamics in patients with EH (allostasis or homeostasis). Due to the preservation of allostatic control mechanisms in patients receiving antihypertensive therapy that reached target values, there is doubt whether we can consider this treatment as pathogenetic. The features of ALI formation in patients in groups 1 and 2 allow us to identify directions for searching additional therapeutic effects. In particular, for group 1, it is the normalization of the nighttime BP profile. For patients in group 2, it should be the optimization of the activity of the sympathoadrenal system and the activity of the RAAS. However, the possibility of transferring the mechanisms of central hemodynamic regulation from the state of allostasis to homeostasis remains controversial.

## 5. Conclusions

Despite the drug treatment, patients in both groups showed a statistically significant increase in ALI compared to the control group. However, between groups 1 and 2, no significant difference in ALI was found, in spite of the fact that the pharmacotherapy was different. The hemodynamic mechanisms of the increase in ALI were general and consisted of an increase in the mean values of BP and a decrease in the percentage of 24 h rhythm. Among the differences, group 1 was characterized by a disorder of 24 h profile of BP with a decrease in CI for BP, and a significant increase in the percentage of patients with a reduced CI for BP and HR, and a decrease in amplitude of BP. There was also an increase in the proportion of patients with HR above 80 bpm. For the patients in group 2, an expansion of amplitude for BP and HR and an increase in range of oscillations for BP was characteristic, which indicates increased reactivity of the cardiovascular system. Summarizing the results obtained, we could come to the following conclusions:Despite the ongoing antihypertensive therapy, regulation of the cardiovascular system realized within the frame of hemodynamic allostasis in patients with EH without metabolic syndrome.In the absence of metabolic syndrome in patients with EH, the main contribution to ALI can be made by hemodynamic parameters, which differ in their characteristics depending on the nighttime profile of BP. In particular, in group 1, the main role in the formation of allostatic load belongs to the changes in the nighttime profile of BP, and in group 2, the changes in the range of oscillations of BP are more pronounced.The persistence of allostatic load in the patients of both groups, considered to be low in value, allows us to raise a question concerning the effectiveness of pharmacotherapy in relation to allostasis. This approach involves assessing the adequacy of therapy not only by hemodynamic parameters but also using ALI for this purpose.

## Figures and Tables

**Figure 1 diagnostics-13-03553-f001:**
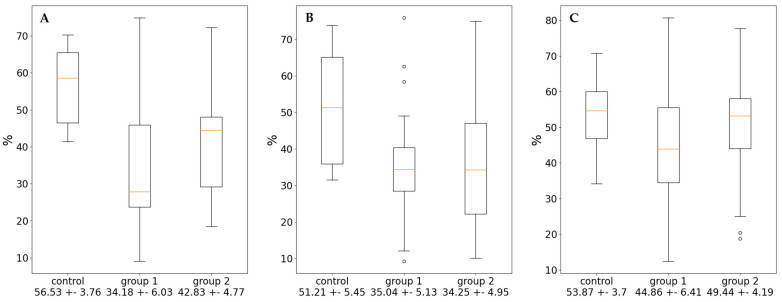
Rhythm power (% of rhythm) of SBP (**A**), DBP (**B**), and HR (**C**).

**Figure 2 diagnostics-13-03553-f002:**
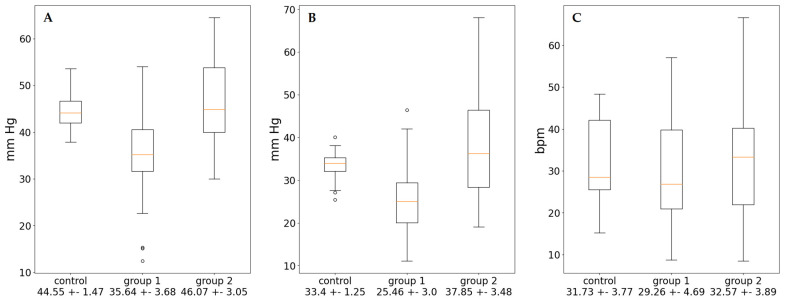
Range of oscillations of SBP (**A**), DBP (**B**), and HR (**C**).

**Table 1 diagnostics-13-03553-t001:** Clinical and laboratory characteristics of the analyzed group of patients.

Indicator	Normal Range	Group 1 (*n* = 32)	Group 2 (*n* = 40)
Age, years	–	58.25 ± 4.05	59.5 ± 3.03
Men, %	–	50	37.5
Women, %	–	50	62.5
BMI, kg/m^2^	–	26.65 ± 0.39	26.85 ± 0.4
WC, cm	–	91.7 ± 1.33	91 ± 1.15
Duration of EH, years	–	12.35 ± 1.2	12.3 ± 1.15
HbA1c, %	4–6.2	6.0	5.7
Creatinine, mcmol/L	64–92	83.9 ± 3.9	82.9 ± 2.11
Urea, mmol/L	3–9	6.25 ± 0.5	5.95 ± 0.42
MAU, mg/L (morning average sample)	˂20	0.6 ± 0.01	0.5 ± 0.002
IRI, mkEd/mL	2–25	26.65 ± 0.39	26.85 ± 0.4
EF, %	55–70	63.7 ± 0.26	63.45 ± 0.16
Patients treated with ACE inhibitors or ARBs, %	–	78.1	82.5
Patients treated with β-blockers, %	–	21.90	12.5
Patients treated with diuretics, %	–	50.00	55
Patients treated with Ca^2+^ antagonists, %	–	21.9	12.5

Note: BMI—body mass index; WC—waist circumference; HbA1c—glycated hemoglobin; MAU—microalbuminuria; IRI—immunoreactive insulin; EF—ejection fraction.

**Table 2 diagnostics-13-03553-t002:** Comparative linear analysis of the 24 h profiles of hemodynamic parameters.

Indicator	Control (*n* = 30)	Group 1 (*n* = 32)	Group 2 (*n* = 40)
24 h SBP, mmHg	120 ± 1.87	139.6 ± 2.2 *	134.0 ± 1.47
24 h DBP, mmHg	76.2 ± 1.55	79.5 ± 2.19	78.15 ± 1.17
24 h HR, bpm	76.2 ± 1.80	76.08 ± 2.7	70.45 ± 1.16
CV for SBP	0.06	0.09	0.06
CV for DBP	0.08	0.15	0.09
CV for HR	0.09	0.2	0.1
Daytime SBP, mmHg	121.3 ± 1.85	141.4 ± 2.5 *	140.1± 1.4 *
Nighttime SBP, mmHg	104.6 ± 2.05	142.1 ± 2.4 *	119.75 ± 1.9 *•
Daytime DBP, mmHg	76.8 ± 2.01	78.1± 2.1	78.6 ± 1.81
Nighttime DBP, mmHg	63.6 ± 1.01	76.4 ± 2.5 *	64.7 ± 1.9 •
Daytime HR, bpm	75.9 ± 1.24	73.8 ± 1.82	72.5 ± 1.73
Nighttime HR, bpm	64.3 ± 3.14	63.9 ± 1.75	60.4 ± 0.9
CI for SBP	1.18 ± 0.04	1.00 ± 0.01 *	1.2 ± 0.03 •
CI for DBP	1.2 ± 0.02	1.02 ± 0.02 *	1.2 ± 0.02 •
CI for HR	1.18 ± 0.01	1.15 ± 0.02	1.2 ± 0.02
Time index for SBP, %	22.9 ± 3.21	60.9 ± 5.71 *	57.8 ± 3.9 *
Time index for DBP, %	18.4 ± 2.7	42.7± 5.8 *	38.4 ± 2.91 *
DP	89.6 ± 0.27 (moderate)	99.5 ± 2.40 (low)	97.0 ± 1.9 (low)
SPBP	0.62 ± 0.002 (100%)	0.54 ± 0.08 * (87%)	0.57 ± 0.07 * (92%)

Note: CV—coefficient of variation; CI—circadian index; DP—double product; SPBP—structural point of BP. *p* ≤ 0.05 •—in comparison with group 1; *—in comparison with the control group.

**Table 3 diagnostics-13-03553-t003:** Non-linear analysis of 24 h profiles of BP and HR.

Indicator	Control (*n* = 30)	Group 1 (*n* = 32)	Group 2 (*n* = 40)
Mesor			
SBP, mmHg	114.36 ± 1	140.3 ± 0.14 *	135.2 ± 1.37 *•
DBP, mmHg	71.15 ± 1.52	77.9 ± 0.42 *	78.0 ± 1.73 *
HR, bpm	72.16 ± 1.05	75 ± 1.9	74.4 ± 1.13
**Amplitude**			
SBP, mmHg	18.06 ± 1.6	21.5 ± 0.99	26.8 ± 0.28 *•
DBP, mmHg	15.5 ± 0.95	13.65 ± 0.31	20.55 ± 0.6 *•
HR, bpm	16.31 ± 0.9	16.03 ± 0.01	21.5 ± 0.56 *•

Note: *p* ≤ 0.05 •—in comparison with group 1; *—in comparison with the control group.

**Table 4 diagnostics-13-03553-t004:** Clinical and laboratory features and ALI.

Indicator	Control (*n* = 30)	Group 1 (*n* = 32)	Group 2 (*n* = 40)
SBP ≥ 120 mmHg, % of patients	30	100 *	100 *
DBP ≥ 80 mmHg, % of patients	27	37.5	40
HR ≥ 80 bpm, % of patients	30	100 *	100 *
CI for 3 indicators ≤ 1, % of patients	0	21.8 *	0 ^×^
Range SBP ≥ 45 mmHg, % of patients	0	10	41.5 ^×^*
Range DBP ≥ 33 mmHg, % of patients	0	18.8	63.4 ^×^*
Range HR ≥ 32 b/m, % of patients	0	37.5 *	48.7 *
ALI			
0	73.3 *	3.1	5
1–2	26.7	87.5 *	90 *
3–4	0	9.3	5
5 or above	0	0	0

Note: In the analysis (range, % of patients) the values of the control group are taken. Fisher angular transformation for percentages (one-way test) was applied. CI for 3 indicators: reduction in CI less than 1 for SBP, DBP, and HR at a time. *p* ≤ 0.05 ^×^—in comparison with group 1; *—in comparison with group 2.

## Data Availability

Data are contained within this article.

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
