# Peer review of "Features of Allostatic Load in Patients with Essential Hypertension without Metabolic Syndrome Depending on the Nature of Nighttime Decreases in Blood Pressure"

_diagnostics, 2023, doi:10.3390/diagnostics13233553_

Round 1
Reviewer 1 Report
Comments and Suggestions for Authors
Dear Authors
I read and checked your article carefully. The title is very clear, the abstract is sufficient, although the introduction part is short, it gives enough information and the main purpose of the study is mentioned. The inclusion criteria and exclusion criteria are mentioned. The findings are clearly explained, the tables are explanatory. The discussion section can be expanded, especially if the first sentence and the first paragraph of the discussion start with the main results of your study, I think it would be more elegant. As a result of the evaluation, no difference was found between groups 1 and 2 in terms of ALI, compared to the control group, which can be expected since the control group did not have a diagnosis of HT. While comparing the groups, blood pressure medications could have been mentioned in more detail. Hormonal mechanisms may differ according to individuals. The scope of the study could be expanded and the number of patients could be increased.
Reviewer 2 Report
Comments and Suggestions for Authors
Overview of the manuscript
The manuscript focuses on the evaluation of allostatic load (ALI) in patients suffering of essential hypertension without metabolic syndrome.
The authors monitored two group of patients, one without significant nighttime decrease in blood pressure (BP) another with nocturnal decrease in BP, compared to the control group of healthy subjects. Linear and non-linear rhythm analyses were applied to evaluate several hemodynamic and metabolic parameters.
The authors found that ALI was significantly higher in groups 1 and 2 compared to the control group, and hemodynamic mechanisms of this increase were different. In conclusion the authors highlighted the persistence of the allostatic load in patients treated with antihypertensive therapy, raising a question concerning the effectiveness of pharmacotherapy in relation to allostasis.
GENERAL COMMENT
The work is very interesting and adds something news to the problem of controlling the hypertensive level, expressed by parameters of hemodynamic allostasis and allostatic load on which the authors are experts as evidenced by several publication on this topic. The manuscript is well performed and rich in the determination and presentation of data that agree with the author’s conclusions. The statistical analysis is adequate and rigorous. Some points should be revised for better readability of the manuscript.
SPECIFIC COMMENTS
Materials and Methods
Pag. 2, line 72-73: the sentence is not clear. Rephrase it.
Pag. 4, line 146-152: the paragraph should be shifted to the beginning of the Results section
Results
Pag. 6, Table 3: the significance is put in the wrong place.
Pag. 7, line 230: in materials and methods section ALI (3-4) has been indicated as moderate and not elevate. Change or explain it.
Pag. 7, Table 4: the ALI level 5 is not reported. It should be indicated, even if 0 is its values.
In the Table 4 the significance indicators are not comprehensible.
Round 2
Reviewer 1 Report
Comments and Suggestions for Authors
Dear Authors
As a result of the reviewers' criticisms and suggestions, many parts of the article have been added and removed. In its current form, it has become a more intriguing and higher quality article. As such, I do not have any additional suggestions.